| 1                 | Documenting the 2015-2017 freshening of the eastern Eurasian Basin of the                                              |
|-------------------|------------------------------------------------------------------------------------------------------------------------|
| 2                 | Arctic Ocean and evaluating its drivers and consequences                                                               |
| 3                 |                                                                                                                        |
| 4                 | Dolly More <sup>1</sup> and Igor V. Polyakov <sup>1</sup>                                                              |
| 5                 | <sup>1</sup> International Arctic Research Center and College of Natural Science and Mathematics, University of Alaska |
| 6                 | Fairbanks, Fairbanks, AK 99775, USA.                                                                                   |
| 7<br>8<br>9<br>10 | Corresponding author: Igor Polyakov (ivpolyakov@alaska.edu)                                                            |

https://doi.org/10.5194/egusphere-2025-5592 Preprint. Discussion started: 19 November 2025 © Author(s) 2025. CC BY 4.0 License.

11 Abstract 12 The Arctic Ocean is undergoing rapid change, with freshwater playing a central role in shaping stratification, 13 vertical heat exchange, and sea-ice loss. Using long-term observations from the Nansen and Amundsen Basins 14 Observational System (NABOS), we document an extreme freshening event in the eastern Eurasian Basin 15 between late 2015 and early 2017. During this period, salinity in the upper 175 m decreased by ~0.5 psu, 16 equivalent to an additional ~0.6 m of freshwater, relative to the preceding (2013-2015) and following (2017-17 2018) years. The anomaly originated on the Kara Sea shelves in 2014-2015, when exceptional Yenisey and Ob 18 discharge provided a combined freshwater surplus of ~0.78 m, sufficient to explain the observed freshening. 19 Trajectory analysis traced the freshwater anomaly to the Kara Sea, with transport times of 8-9 months to the 20 shelf and 22-23 months to offshore. The resulting enhanced stratification suppressed upper-ocean currents by 21  $\sim$ 22% and vertical shear by  $\sim$ 50%, reducing vertical heat flux from the ocean interior. These changes enabled 22 thicker sea ice to persist through the summers of 2016-2017, in contrast to near-ice-free conditions in adjacent 23 years. While wind anomalies aided the retention of freshwater along the slope, river discharge was the dominant 24 driver of the event. Overall, the 2015-2017 event demonstrates how episodic river discharge events can 25 restructure upper-ocean stratification, reduce oceanic heat fluxes, and lead to delayed melt and increased 26 summer sea ice, highlighting the sensitivity of upper-ocean processes and sea ice to episodic freshwater forcing 27 in the Arctic.

1. Introduction

28

29

30

define this transformation. Surface air temperature has increased by ~1.5°C per century since 1901, while the 32 Arctic Ocean interior has warmed by ~0.8°C per century (IPPC, 2021). Sea-ice extent has declined by ~0.3 33 million km² per decade since 1979 (Cavalieri et al., 2003), exemplifying the phenomenon of Arctic 34 amplification (Kim et al., 2017; Overland and Wang, 2016). 35 Among the key consequences of this warming is the increasing input and redistribution of freshwater from 36 rivers, glacial melt, and precipitation (Haine et al., 2015; Alkire et al., 2017). Although the Arctic Ocean covers 37 only a small fraction of the global ocean area, it holds a disproportionately large share of the global ocean's 38 freshwater (Serreze et al., 2006). Model projections suggest that Arctic River discharge could rise by 20-50% 39 by 2100 (Haine et al., 2015; Stadnyk et al., 2021; Rawlins and Karmalkar, 2024). Surface freshwater 40 accumulation strengthens upper-ocean stratification, forming a low-salinity cap that insulates sea ice from 41 underlying Atlantic Water (AW) heat (Carmack et al., 2015; Polyakov et al., 2013, 2020b). Conversely, surface 42 salinification weakens stratification, promotes vertical mixing, and accelerates sea-ice melt. 43 Recent observations highlight the strong control of freshwater on sea-ice variability. In the Canadian Basin, 44 enhanced freshwater since 2007 suppressed oceanic heat fluxes and temporarily slowed summer ice loss in the 45 late 2010s (Polyakov et al., 2023). Such episodes demonstrate how regional freshwater dynamics can modulate 46 or even interrupt long-term sea-ice decline. 47 The Siberian Arctic plays a central role in these processes. About 11% of global river discharge enters the 48 Arctic Ocean (Dai and Trenberth, 2002), and roughly 65% of this input comes from the Ob, Yenisey, and Lena 49 Rivers (Haine et al., 2015; Shiklomanov et al., 2021). These rivers deliver freshwater and dissolved materials 50 that influence stratification, nutrient distribution, and biological productivity on the Siberian shelves (Dittmar 51 and Kattner, 2003; Yamamoto-Kawai et al., 2013). The deep Arctic basins, connected to these shelves, receive

The Arctic is warming faster than any other region on Earth, with profound physical and ecological

consequences. Rising air temperatures, declining sea ice, ocean warming, and intensified hydrological cycles

transport to large-scale freshening events remain poorly constrained (Pemberton et al., 2014; Timmermans and Marshall, 2020; Laukert et al., 2025). Sparse sampling beneath sea ice limits our ability to track freshwater pathways and variability. Models often poorly resolve mesoscale processes that redistribute freshwater (Nguyen et al., 2011; Müller et al., 2024), leading to biases in simulated freshwater storage and export (Hoffman et al., 2023). Moreover, changes in river discharge timing and snowmelt patterns further complicate predictions of freshwater release and retention (Rawlins et al., 2010; Stroeve et al., 2014).

much of this freshwater through complex interactions among shelf-basin exchange, wind forcing, and the AW

Despite growing observations, the relative contributions of river discharge, shelf storage, and cross-slope

boundary current (Aksenov et al., 2011; Pnyushkov et al., 2015; Baumann et al., 2018).

This study addresses these gaps by examining the 2015–2017 extreme freshening event in the eastern Eurasian Basin. The objectives are to (1) quantify how riverine freshwater from the Yenisey and Ob Rivers contributed to this event, (2) investigate its pathways, transport timescales, and shelf–slope retention processes, and (3) assess its impacts on upper-ocean stratification, halocline structure, and sea-ice persistence. Using long-term mooring records, hydrographic observations, and reanalysis products, we evaluate how enhanced stratification during this event suppressed vertical shear and reduced upward heat flux from AW, enabling thicker and longer-lasting sea ice. By linking freshwater sources to their physical impacts, this study provides

new insight into how episodic river-driven freshening modulates upper-ocean structure and sea-ice variability in the Eurasian Basin.

## 70 2. Data and Methods

2.1 Oceanic data

a) Mooring observations

Our analysis uses records from six moorings (M1<sub>1</sub>, M1<sub>2</sub>, M1<sub>3</sub>, M1<sub>4</sub>, M1<sub>5</sub>, and M3) deployed in the eastern Eurasian Basin, spanning the continental slope from the shallower near-shelf region to the deeper offshore slope between September 2013 and September 2018 (Fig. 1; Table 1). Together, these moorings sampled depths from approximately 30 m to 2700 m (depth ranges for individual moorings are listed in Table 1). Most moorings conducted CTD (Conductivity–Temperature–Depth) observations using fixed-depth SBE37 microcat instruments, complemented by SBE56 thermistors. Two moorings (M1<sub>3</sub> and M1<sub>5</sub>) were equipped with McLane Moored Profilers (MMPs; Table 1), which measured vertical profiles of temperature, salinity, and currents daily at a profiling speed of ~25 cm s<sup>-1</sup>, achieving ~12 cm vertical spacing with a 0.5 s sampling interval. The MMPs provided temperature and conductivity measurements with calibration accuracies of  $\pm 0.002$  °C and  $\pm 0.002$  mS cm<sup>-1</sup>, respectively. The SBE37 recorded data every 15 minutes with accuracies of  $\pm 0.002$  °C for temperature and  $\pm 0.003$  mS cm<sup>-1</sup> for conductivity. The SBE56 temperature sensors had an accuracy of  $\pm 0.002$  °C.

Figure 1: Bathymetry map of the eastern Eurasian Basin (shaded by depth, in meters) showing the locations of six NABOS moorings (yellow circles). The inset in the upper right shows the Arctic Ocean, with the Laptev Sea region outlined by a black box.

Table 1: Summary of the NABOS mooring data used in this study, including instrument details. For mooring locations, see Fig. 1.

| Moorings         | Latitude (°N),<br>Longitude (°E) | Depth (m) | Instrument(s   | Depth range (m)                                           | Beginning of record | End of record |
|------------------|----------------------------------|-----------|----------------|-----------------------------------------------------------|---------------------|---------------|
| M1 <sub>1a</sub> | 77, 04.221<br>125, 49.577        | 250       | SBE37          | 53, 140, 240                                              | 26.08.2013          | 10 .09. 2015  |
| M1 <sub>1b</sub> | 77, 04.221<br>125, 49.577        | 252       | SBE37          | 53, 140, 240                                              | 21.09.2015          | 3 .09. 2018   |
| M1 <sub>2a</sub> | 77, 10.373<br>125, 47.974        | 787       | SBE37          | 70-754<br>53                                              | 26 .08. 2013        | 31 .08. 2015  |
| M1 <sub>2b</sub> | 77, 10.373<br>125, 47.974        | 783       | SBE37<br>SBE56 | 31,44,67,138,2<br>13,266,628<br>34, 37, 40, 47,<br>50, 53 | 21 .09. 2015        | 3 .09. 2018   |
| M1 <sub>3a</sub> | 77, 39.234<br>125, 48.686        | 1849      | MMP            | 64-750                                                    | 7 .09. 2013         | 3 .09. 2015   |
| M1 <sub>3b</sub> | 77, 39.234<br>125, 48.686        | 1866      | MMP            | 70-1056                                                   | 22 .09. 2015        | 15 .06. 2017  |
| M1 <sub>4a</sub> | 78, 28.084<br>125, 57.679        | 2721      | SBE37          | 62, 129, 214,<br>265, 617                                 | 5 .09. 2013         | 19 .09. 2015  |
| M1 <sub>4b</sub> | 78, 28.084<br>125, 57.679        | 2700      | SBE37          | 38, 107, 188,<br>240, 604                                 | 21 .09. 2015        | 18 .09. 2018  |
| M1 <sub>5a</sub> | 79, 56.194<br>126, 01.228        | 3443      | MMP            | 88-754                                                    | 28 .08. 2013        | 21 .08. 2015  |
| M1 <sub>5b</sub> | 79, 56.194<br>126, 01.228        | 3443      | MMP            | 172-806                                                   | 24 .09. 2015        | 29 .08. 2018  |
| M3e              | 79, 56.136<br>142, 14.887        | 1335      | SBE37          | 41, 45, 57, 64,<br>130, 270, 600                          | 31 .08. 2013        | 7 .09. 2015   |
| M3f              | 79, 56.194<br>142, 15.216        | 1357      | SBE37          | 30, 50, 133,<br>217, 268, 614                             | 7 .09. 2015         | 6 .09. 2018   |

90

To evaluate the potential effect of using fixed-depth SBE37 data, we performed a sensitivity test by subsampling the high-resolution MMP salinity data at typical SBE depths and recalculating freshwater content over the full MMP depth range. Differences between the original and subsampled freshwater estimates were less than 1%, indicating that interpolation of SBE data introduces negligible error.

939495

Current measurements were obtained using 300 kHz Acoustic Doppler Current Profilers (ADCPs) and MMPs. ADCPs measured velocity in 2 m vertical bins with at least 1-hour time resolution and an accuracy of 0.5% of measured speed and 2 degrees for direction. MMPs profiled from  $\sim$ 40 to 1000 m every two days at  $\sim$ 25 cm/s with a 0.5-second sampling rate, yielding vertical spacing of  $\sim$ 12 cm. The acoustic current meters on MMPs have a velocity error of  $\pm$ 0.5 cm/s. Directional accuracy of the MMP compass is 2 degrees, but in the Arctic, errors can reach up to 30 degrees due to weak horizontal geomagnetic fields (Thurnherr et al., 2017).

98 99

For this study, we merged data from two consecutive mooring deployments (2013–2015 and 2015–2018) to

100101

create continuous temperature and salinity time series on a unified 2 m vertical grid using linear interpolation.

All eastern Eurasian Basin mooring data were collected and made publicly available by the Nansen and

Amundsen Basins Observational System (NABOS) at the University of Alaska Fairbanks (https://uaf-

b) Water mass fraction data

iarc.org/nabos/data/).

Concentrations of meteoric water (MW) and sea-ice meltwater (SIM) fractions were derived from NABOS 107 macronutrient observations described in Polyakov et al., (2020a) and available through the Arctic Data Center 108 (Whitmore et al., 2023). The dataset includes discrete measurements of oxygen isotope composition ( $\delta^{18}O$ ) and 109 salinity collected during oceanographic cruises across the Arctic Ocean between 1981 and 2017, primarily 110 during May-October at latitudes north of 60°N. Seawater samples were obtained using Niskin bottles at selected 111 depths, and  $\delta^{18}$ O and salinity were analyzed to calculate MW and SIM fractions. Temperature, salinity, and 112 oxygen data from accompanying CTD profiles were matched to sampling depths, and discrete measurements 113 were used to validate sensor accuracy. In this study, we used MW and SIM fraction data from 2013 and 2015 to 114 examine changes in freshwater composition in the Kara and Laptev Seas preceding and during the onset of the 115 2015-2017 freshening event. 116 2.2. Sea ice concentration 117 Sea ice concentration data were obtained from the Advanced Very High-Resolution Radiometer satellite 118 archive, which provides global daily sea ice concentration from 1981 to 2021 at 0.25°×0.25° spatial resolution 119 (Comiso, 2017). For this study, we used the data corresponding to the mooring deployment period (September 120 2013 to September 2018). Sea ice concentration values corresponding to each mooring site were obtained by 121 sampling the dataset at the latitude and longitude coordinates listed in Table 1. 122 2.3. River discharge data 123 Daily discharge data (m³ s-1) for the Yenisey, Ob, and Lena Rivers were obtained from the Arctic Great Rivers 124 Observatory (ArcticGRO) archive (discharge product version 20220630; https://arcticgreatrivers.org/discharge). 125 The dataset, compiled by Russia's national hydrological agency (Roshydromet), covers the period 1979-2020 at 126 daily resolution (Holmes et al., 2022). Published assessments indicate annual mean discharge uncertainties of 127 approximately 1.5-3.5% for large Siberian rivers (Shiklomanov et al., 2006), with higher reliability during the 128 open-water season. To minimize uncertainty, only May-October discharge data were used to estimate seasonal 129 totals and analyze interannual variability (Section 5.2). 130 2.4. Reanalyses data 131 a) Ocean ORAS5 reanalysis 132 We used monthly salinity and ocean current data from the Ocean Reanalysis System 5 (ORAS5) and its near-133 real-time extension, Ocean5 (Zuo et al., 2019), produced by the European Centre for Medium-Range Weather 134 Forecasts (ECMWF). The ORAS5 product has a horizontal resolution of approximately 0.25° × 0.25° (about 25 135 km in the tropics and 9 km in the Arctic) and spans from 1979 to the present. It is generated using the NEMO 136 v3.4 ocean model with the NEMOvar data assimilation system, which employs a 3D-Var FGAT (First Guess at 137 Appropriate Time) scheme and assimilates both satellite and in situ observations to constrain ocean dynamics. 138 For salinity, we used monthly mean values from the upper 0-5 m layer to examine surface salinity variability in 139 the Kara and Laptev Seas. For currents, we used monthly averaged zonal and meridional velocity components 140 from the same layer for January 2011-December 2018 in a back-trajectory tracer analysis to investigate 141 freshwater transport pathways. 142 b) Atmospheric reanalysis data 143 Monthly surface wind fields from 2013 to 2018 were obtained from the ERA5 reanalysis, available via the 144 Copernicus Climate Data Store (https://cds.climate.copernicus.eu). The dataset provides 10-meter wind

## https://doi.org/10.5194/egusphere-2025-5592 Preprint. Discussion started: 19 November 2025 © Author(s) 2025. CC BY 4.0 License.

components at a horizontal resolution of 0.25° × 0.25° (Hersbach et al., 2020). These wind fields were used to evaluate the role of atmospheric forcing in the 2015–2017 freshening event in the eastern Eurasian Basin.

2.5. Defining the timing and vertical extent of the freshening event

a) Defining the timing of freshening events using wavelet

To determine the timing of the freshening event, we applied wavelet analysis to a salinity time series for each mooring location (Torrence and Compo, 1998). The study used DOG (derivative of the Gaussian) mother wavelet with 95% confidence intervals. Wavelet transforms were computed using a standard wavelet analysis package to isolate seasonal-scale variability and identify statistically significant low-salinity anomalies associated with the freshening event (Fig. S1). Based on these results, the event was defined as occurring from late 2015 to early 2017, with slight variations in onset and termination across mooring sites. Equal-duration reference periods before and after the event were selected for comparative analysis. The identified freshening period represents an anomalous decrease in salinity across the upper halocline layer, clearly resolved in the 65–100 m salinity wavelet signals (Fig. S1).

b) Defining the freshening layer

To define the vertical extent of the freshening layer, we computed a mean salinity anomaly profile at each mooring by subtracting the average salinity profiles from before and after the event from that during the event (Fig. S3). The upper limit of the freshening layer,  $z_1$ , corresponds to the shallowest reliable salinity record at each mooring. Two lower limits were identified from the salinity profile. The first,  $z_2$ , is the depth where the anomaly amplitude decreased to 3% of its surface value (6% for M1<sub>5</sub>). The second,  $z_2$ ', marks the depth where the anomaly became statistically indistinguishable from zero ( $\leq 0.003$  psu, or 3% of the surface anomaly for M1<sub>5</sub>; Table S1).

To evaluate the sensitivity of results to layer definition, freshwater content was integrated between  $z_1$  and each of the lower limits  $z_2$  and  $z_2$ ' (Fig. S4). The difference in mean freshwater content between the two thresholds was less than 3%, and the two timeseries were strongly correlated ( $r \ge 0.99$ ) across all moorings, indicating minimal sensitivity to the choice of lower boundary. Based on this result, the shallower limit  $z_2$  was used for further analysis (Fig. S3). For M1<sub>5</sub>, wavelet analysis showed that the freshening signal became statistically significant below 75 m for the period before the event, supporting the use of 77 m as the upper limit ( $z_1$ ) for defining the freshening layer.

- 2.6. Quantifying ocean responses
- a) Quantifying freshwater and heat content

Freshwater content was calculated by integrating salinity anomalies over depth, following the approach of Aagaard and Carmack, (1989).

Freshwater content =  $\int_{z1}^{z2} \frac{1}{S_0} (S_z - S_0) dz$

where  $S_0 = 34.8$  psu used as reference salinity,  $z_1$  and  $z_2$  are depths of upper and lower boundaries, dz is the ocean layer thickness, and  $S_z$  is the observed salinity at depth z.

Ocean heat content (J/m²) was calculated using temperature with the freezing point as the reference 182 temperature at a given salinity (Polyakov et al., 2017).

$$Q = \int_{z_1}^{z_2} C_p \, \rho \left( T - \theta_f \right) dz$$

- where  $\theta_f$  is the freezing temperature,  $(-0.054*S_z$ , with  $S_z$  the observed salinity at depth z, which is used as a 185 186 good proxy for the seawater freezing point),  $\rho$  is water density,  $C_p$  is the specific heat of seawater, and  $z_1$  and  $z_2$ 187 are depths of upper and lower boundaries. Here, Q can be defined as the relative heat content, which measures 188 the amount of heat that must be removed to create ice crystals at a given salinity and pressure.
- b) Available potential energy
- The available potential energy (APE) provides a measure of stratification and is used to quantify the strength 191 of the surface mixed layer and halocline. APE was calculated following (Polyakov et al., 2018), by integrating 192 the density anomaly relative to a reference from the surface to the base of the freshening layer, using the 193 formula below.

$$APE = \int_{z_1}^{z_2} g(\rho_i - \rho_{ref}) z \, dz$$

- where  $z_1$  is the surface,  $z_2$  is the depth of the freshening layer, g is gravitational acceleration,  $\rho_i$  is the density of 196 the water at a given depth,  $\rho_{ref}$  is the reference density (a constant depth density used for comparison), and z is 197 the ocean depth.
- 2.8. Identifying freshwater sources and pathways
- a) Cross-slope salinity shifts across the Laptev Sea slope
- To assess whether cross-slope shifts of the Atlantic core of the boundary current could cause freshening 201 across the eastern Eurasian Basin slope, we defined a proxy of the advective term between pairs of adjacent 202 moorings (M1<sub>1</sub>-M1<sub>5</sub>). For each mooring, daily salinity and meridional velocity profiles were averaged over the 203 50-150 m layer from September 2015 to September 2018 (excluding the freshening period from October 2015 204 to September 2017). For each set of two adjacent moorings, the cross-slope salinity difference  $\Delta S$  and mean 205 meridional velocity  $\vec{V}$  was computed. We used the product of the mean meridional velocity  $(\vec{V})$  between the 206 same two moorings and the salinity difference ( $\Delta S$ ) as a proxy of the advective term ( $\bar{V}(\Delta S)$ ). A positive value of 207 the proxy indicates potential salinification due to advection at the northern mooring, i.e., saltier water moving 208 northward, while a negative value indicates freshening there.
- b) Defining connections between Kara and Laptev seas salinity

timing of salinity signal propagation between the two seas.

- Deseasoned monthly surface salinity time series from the Laptev and Kara Seas were compared using cross-211 correlation analysis to assess temporal relationships in salinity variability. The Kara Sea region was selected for 212 this comparison based on back-trajectory results indicating the primary pathway of Yenisey and Ob River 213 discharge through the Kara Sea. The lag corresponding to the maximum correlation was used to determine the 214
- c) Tracer trajectory calculation

To investigate the source of freshening in the eastern Eurasian Basin, we used a Lagrangian trajectory model 217 developed by A. Pnyushkov and used in Polyakov et al., (2023). The trajectories were computed using the 218 ORAS5 monthly velocity field, averaged over the 0-5 m depth layer, for the period 2013-2018. Parcels were 219 initialized at each mooring position and integrated backward in time for 23 months, which corresponds to the 220 maximum lag found between Kara and Laptev seas salinity anomalies (see Section 3.2). All tracers were 221 assumed to have neutral buoyancy and were advected at a constant depth over the duration of their drift from the 222 initialization point. 223 3. Results 224 3.1 Documenting the 2015-2017 freshening event 225 An extreme freshening event was observed across six mooring locations (M11 to M15, and M3) in the eastern 226 Eurasian Basin from 2015 to 2017 (Fig. 2). At all these locations, the vertical structure of the salinity anomaly 227 showed that the surface layers had the strongest freshening (up to 0.5 psu), and the freshening extended to 228 depths up to 175 m (except for the M1<sub>1</sub> mooring location, where the anomaly extended deeper, to 220m; see 229 Methods). The evolution of salinity and temperature anomalies from September 2013 to September 2018 is 230 illustrated using a Hovmöller diagram (Fig. 3). For this purpose, salinity and temperature anomalies at 77 m, the 231 shallowest level with continuous records with a few gaps across all moorings, were analyzed. Low salinity 232 (fresher) anomalies (up to -0.4 psu) first appeared at the shallower moorings M1<sub>1</sub>-M1<sub>3</sub> in late 2015, and were 233 observed later at the offshore moorings M1<sub>4</sub>–M1<sub>5</sub> and M3 by early 2016 (Figs. 3a and 3b). These anomalies 234 added nearly 0.60 m of freshwater to the upper ocean layer (~ 30 m to 175 m), with freshwater content 235 increasing from winter 2015 and reaching a maximum in 2016 at M12 and M13, followed by a gradual decline 236 observed at all moorings after 2017 (Fig. 4). These anomalies persisted until early 2017 across all mooring sites. 237 The added freshwater reduced near-surface salinity and density, thereby strengthening the vertical density 238 gradient. This enhanced stratification was reflected in the pronounced increase in estimated available potential 239 energy across all moorings (Fig. 5). For example, potential energy nearly doubled at M1<sub>1</sub> (from 0.66 x10<sup>4</sup> to 240  $1.24 \times 10^4 \text{ J/m}^2$ ), increased by ~70% at M1<sub>3</sub> (from 0.90  $\times 10^4 \text{ to } 1.52 \times 10^4 \text{ J/m}^2$ ), and more than doubled at M3 241 (from 1.86 x10<sup>4</sup> to 3.90 x10<sup>4</sup> 10<sup>4</sup> J/m<sup>2</sup>) during the event. This increased stratification is a potential contributor to 242 reduced vertical heat exchange between surface waters and the underlying warm Atlantic layer over the 243 freshening event. 244 The 2015-2017 freshening event spanned over three seasons: two winters (2015-2016, 2016-2017) and one 245 summer (2016). To examine how salinity varied across seasons during the freshening event, we analyzed 246 seasonal averages for winter (DJF) and summer (JJAS) periods. Fig. 6 shows that the event was not uniform 247 across all locations and seasons. On average, salinity decreased by ~0.2 psu during winter 2015-2016, with the 248 strongest anomalies at M3 (0.25 psu) and M14 (0.43 psu) mooring locations. In summer 2016, salinity increased 249 slightly compared to the preceding winter. However, summer 2016 remained ~0.1-0.2 psu fresher than both 250 summer 2015 and summer 2017, so that the freshening persisted over the three seasons despite seasonal 251 variability.

Figure 2: Depth–time distributions of anomalous (relative to the 2013-2018 mean) salinity from six mooring locations in the eastern Eurasian Basin up to 175m (see their positions in Fig. 2.1) from 2013 to 2018. Vertical red dashed lines indicate the freshening event (2015-2017), and solid red lines mark the periods before and after the event, used for comparative analysis. White gaps indicate missing data.

Figure 3: Hovmöller diagrams of temperature (a) and salinity (b) anomalies (relative to the 2013–2018 mean) at 77 m depth for six mooring locations in the eastern Eurasian Basin. The 77m depth was selected based on the shallowest continuous salinity record across all moorings. The horizontal axis represents mooring site labels, and the vertical axis shows time (September 2013 to September 2018). Horizontal black lines mark the start of each year. White gaps indicate missing data.

Figure 4: Time series of freshwater content for six mooring locations in the eastern Eurasian Basin, averaged over the depth indicated for each mooring individually (bottom right). Horizontal dashed lines represent the mean freshwater content, and the error bar denotes ± three standard errors. Vertical dark blue lines indicate the period of the freshening event; the vertical black lines mark the periods before and after the event.

Figure 5: Time series of available potential energy (APE [ J/m<sup>2</sup>], blue) for six mooring locations in the eastern Eurasian Basin, averaged over the depth indicated for each mooring individually (top right). Horizontal dashed lines represent the mean APE, and the error bar denotes  $\pm 3$  standard errors. Vertical dark blue lines indicate the period of the freshening event; the vertical black lines mark the periods before and after the event.

Figure 6: Mean seasonal salinity averaged for summer (JJAS, orange) and winter (DJF, blue) from 2014 to 2018 from six mooring locations in the eastern Eurasian Basin. Salinity is averaged over the freshening depth identified individually for each mooring record, as indicated in the right panels. Black error bars denote  $\pm 3$ standard errors.

The salinity anomalies were accompanied by positive (warmer) temperature anomalies of up to  $\pm 0.4$  °C at the shallower moorings M1<sub>1</sub>–M1<sub>3</sub> and M3, while deeper offshore moorings M1<sub>4</sub> and M1<sub>5</sub> exhibited mixed temperature signals, lacking a clear tendency toward warming or cooling (Figs. 3a and S4). These anomalies persisted until early 2017 across all mooring sites. During the 2015–2017 freshening event, normalized freshwater content and ocean heat content exhibited contrasting tendencies across mooring locations (Fig. 7). The normalized values are unitless and represent the number of standard deviations ( $\sigma$ ) above or below the mean. At the shallower moorings M1<sub>1</sub> and M1<sub>2</sub>, freshwater content during the event increased by an average of 0.52m, while ocean heat content increased by 0.56 x 10<sup>7</sup> J/m<sup>2</sup>. In contrast, at deeper moorings (M1<sub>3</sub> to M1<sub>5</sub> and M3), the freshwater content and the ocean heat content showed a strong opposite tendency. For instance, the freshwater content averaged over four deeper moorings increased by 0.67m, but ocean heat content declined by 0.60 x 10<sup>7</sup> J/m<sup>2</sup>. The potential underlying mechanisms for these contrasting signals are discussed in Section 6.

Figure 7: Normalized time series of freshwater content (FWC, m; blue) and ocean heat content (OHC,  $J/m^2$ ; orange) averaged over the depth range (indicated in the bottom left of each panel) for six mooring locations in the eastern Eurasian Basin. FWC values are multiplied by -1 so that freshening appears as a negative anomaly, aligning visually with changes in OHC. Horizontal dashed lines represent the mean values for each period. Mean  $\pm$  3 standard errors are shown in dark blue for FWC and in red for OHC. Vertical dark blue lines indicate the period of the freshening event; the vertical black lines mark the periods before and after the event.

295 3.2 Sources and drivers of the 2015-2017 freshening 296 3.2.1 Impact of cross-slope advection, precipitation, and ice melt 297 Variability in the position and strength of the front and boundary currents relative to the slope can 298 significantly affect cross-slope exchanges of water masses, potentially influencing local salinity distributions 299 (Pnyushkov et al., 2015). To evaluate the potential effects of cross-slope shifts of the boundary flow on the 300 2015-2017 freshening, we examined the salinity distribution and the meridional component of the current within 301 the 50-150m layer using data from moorings M1<sub>1</sub> to M1<sub>5</sub> (see Methods). Salinity averaged from September 302 2015 to September 2018 (excluding the freshening period) shows a maximum of 34.55 psu at the mid-slope 303 mooring M13, where the AW core is located, with lower salinities of 34.37 psu at shallower moorings M11-M12 304 and 34.50 psu at outer-slope moorings M14-M1s (Table S2; Fig. S5). To translate these differences into a proxy 305 of an advective term, we used the product of the mean meridional current velocity averaged over the adjacent 306 mooring locations and the meridional salinity difference between these locations (see Methods for details). The 307 meridional flow in the basin is predominantly northward. Consequently, the negative product of the advective 308 proxy indicates northward advective flux of saltier water, implying salification at all moorings except M13 309 (Table S2). However, observations in 2015-2017 show freshening at all mooring locations. Thus, the 2015-2017 310 freshening cannot be explained by the advective cross-slope shift of the salty jet's core. 311 Therefore, processes other than cross-slope shifts of the AW core must have contributed to the anomalous 312 freshening of the upper eastern Eurasian Basin in 2015-2017. Potential contributors are enhanced sea-ice melt, 313 net precipitation, and increased inflow of river-derived meteoric water. Annual precipitation plays a minimal 314 role in the Eurasian Basin, as it accounts for a small portion of the Arctic's yearly freshwater input (Serreze et 315 al., 2006). Additionally, the increasing sea-ice melt is unlikely to explain the 2015-2017 freshening, since it has 316 a relatively low impact on net freshwater content in the eastern Eurasian Basin (Bauch et al., 1995), as reflected 317 by the negative values of sea-ice melt fraction (blue profiles) across all mooring locations (approximately -0.01 318 to -0.02; Fig. 8). Meteoric water (primarily from riverine discharge), on the other hand, is the dominant 319 contributor to freshwater content in the Laptev Sea and eastern Eurasian Basin as shown by the positive 320 meteoric water fraction in the upper 0-50 m (Fig. 8; Bauch et al., 2013; Osadchiev et al., 2024). Notably, in 321 2015, the meteoric water fraction at moorings M1<sub>1</sub> and M1<sub>4</sub> doubled to 0.12% from 0.06% in 2013, whereas the 322 sea-ice-melt fraction remained unchanged, suggesting that anomalous Siberian River discharge may be the 323 cause of the observed freshening in the eastern Eurasian Basin (Fig. 8). 324 3.2.2 Kara Sea as a driver of salinity change 325 Checking records of the peak Siberian River discharges (May-October), we indeed found that the Yenisey 326 and Ob rivers showed an anomalously high runoff peaking between 2014 and 2015 relative to 2013 (Fig. 9). In 327 2014, the Yenisey delivered +6.4 x 10<sup>-6</sup> km<sup>3</sup>/s, an excess equivalent to ~0.19m of freshwater when spread over 328 the area covered by mooring observations (roughly 77-82°N and 110-140°E; 5 x 10<sup>5</sup> km²), while the Ob 329 delivered  $+5.3 \times 10^{-6} \text{ km}^3/\text{s}$ , adding  $\sim 0.16 \text{m}$ . In 2015, the Yenisey supplied  $4.54 \times 10^{-6} \text{ km}^3/\text{s}$  ( $\sim 0.14 \text{m}$ ) and the

332333

increase averaged over 2015-2017 (Fig. 4).

330

331

Ob 8.48 x 10<sup>-6</sup> km<sup>3</sup>/s (~0.26m). Combined, these anomalous ~0.78m discharges from Yenisey and Ob over two

years of 2014-2015 alone can account for the derived from the mooring records ~0.60 m freshwater content

Figure 8: Vertical profiles of sea ice melt (SIM; blue) and meteoric water (MW; orange) fractions at  $M1_1$  and  $M1_4$  mooring locations in the eastern Eurasian Basin for 2013 (lighter shade) and 2015 (darker shade).

Figure 9: Time series of river discharge from the Lena (green), Yenisey (blue), and Ob (orange) Rivers from 2013 to 2018. Dashed black lines indicate the period of the freshening event in the Eurasian Basin. The purple ink circle highlights the increased freshwater discharge from the Lena and Yenisey Rivers before the event. Data source: The Arctic Great Rivers Observatory: https://arcticgreatrivers.org/.

The downstream impact of this anomalous freshwater input is evident in the progressive salinity decline observed across the Kara and Laptev Sea shelf and slope regions (Fig. 10). De-seasoned ORAS5 salinity anomalies, averaged over four-month seasons, reveal a basin-wide freshening that begins in the Kara Sea as early as spring (MAM) 2015 and extends into the Laptev Sea through fall (SON) 2016, indicating the propagation of freshwater from the Kara to the Laptev Sea. This pattern of salinity anomalies highlights the linkage between enhanced Siberian River discharge and the observed freshening in the eastern Eurasian Basin.

Figure 10: Deseasoned ORAS5 surface (1 m depth) salinity anomalies across the Kara and Laptev seas region averaged over seasonal periods (December-February [DJF], March-May [MAM], June-August [JJA], September-November [SON]) from 2013 to 2018.

The cross-correlation analysis of de-seasoned time series of salinity anomalies from Kara and Laptev Sea provides further support to the findings above, showing a positive correlation with values ranging from R = 0.39 at M1<sub>1</sub> to R = 0.78 for M1<sub>5</sub> (all statistically significant at 95% level, Fig. 11). It also indicates that freshwater transport takes between 8 and 23 months to reach the Laptev Sea from the Kara Sea. The lag varied from 7 to 9 months at M1<sub>1</sub> and M1<sub>2</sub> to 22 to 23 months at M1<sub>3</sub>-M1<sub>5</sub> and M3, indicating that freshwater first reached the

shelf edge regions and then the offshore deeper slope areas. This pattern aligns with the different start dates of freshening at mooring locations shown in Fig. 3.

Figure 11: (a) Monthly time series of salinity anomalies averaged across the depth range (sub-surface to 175m) for the Laptev Sea (orange), derived from mooring records, and the Kara Sea (blue), obtained from ORAS5 1m surface salinity (the latter is lagged by maximum correlation). (b) The Kara Sea region (red box) was used to compute the Kara Sea time series. The region was selected based on trajectory analysis, which identified the source region for the Laptev Sea freshening event in 2016-2017.

100°E

125°E

50°E

360

75°E

69°N

150°E

The 2015–2017 freshening event in the eastern Eurasian Basin coincided with changes in wind conditions. Winds from ERA5 reanalysis show pronounced changes over the Kara and Laptev seas during this event, compared to before and after the event periods (Fig. 12). Before and after the event, winds were predominantly southeasterly, however, during the event, the direction shifted to southwesterly (Fig. 12). These changes can alter the direction of Ekman transport, shifting the accumulation of freshwater along the slope rather than driving it into the basin and contributing to the observed freshening buildup. Thus, the timing and magnitude of the Yenisey and Ob discharges, along with anomalous wind conditions, make them the most plausible drivers of the 2015-2017 freshening observed in the eastern Eurasian Basin.

Figure 12: ERA5 10 m wind vectors averaged over three periods: pre-event (top), during (middle), and post-event (bottom), the 2015–2017 freshening event in the Eurasian Basin.

To explore whether the surplus freshwater from the Yenisey and Ob discharges could reach the Laptev slope moorings under anomalous wind conditions, we conducted a Lagrangian tracer analysis. This involved using surface (0-5m) parcels from monthly ORAS5 reanalysis velocities (see Methods for details). These parcels were initialized at each mooring location and advected backward over 23 months, a duration selected to match the salinity lag observed between Kara and Laptev Sea salinity. The resulting trajectories reveal that upper eastern Eurasian Basin freshening originates in the Kara Sea (Fig. 13). Also, the trajectories show that no parcel originates from the Lena River delta, implying that its runoff did not contribute to the observed freshening. The anomalous discharge from the Kara Sea moves eastward as a coastal current, then enters the 10-20 km wide Vilkitsky Strait, and then flows eastward into the eastern Eurasian Basin, following the path identified in Janout et al., (2015). The trajectories remain very similar across all three periods – before, during, and after the freshening event. A negligible change in the trajectory pattern suggests that the direction of anomalous winds did not play a significant role in the freshwater transport. Instead, the large Yenisey and Ob River discharge between 2014 and 2015 (Fig. 9) was probably the dominant source of the freshening observed in the eastern Eurasian Basin.

3.3. Consequences of the eastern Eurasian Basin freshening on ocean currents and sea ice

The 2015-2017 freshening event led to strong stratification in the upper ocean, as evidenced by an increase in the squared buoyancy frequency ( $N^2$ ) down to 100m from late 2015 to 2017 (Fig. 14a). The enhanced upper ocean salinity gradient can significantly suppress vertical mixing and heat ventilation between the surface and the underlying AW (e.g., Miller, 1976; Polyakov et al., 2025). At the same time, the upper 10m current speed |U| and vertical shear of horizontal current  $U_z$  were significantly suppressed during the same period, with maximum weakening observed in 2017 (Fig. 14b-c). The annual mean current speed decreased from ~7.3 cm/s in 2014-2016 to ~5.7 cm/s (a ~22% decline) in 2017, while the annual vertical shear declined from ~9.2 s<sup>-1</sup> in 2014-2015 to ~4.8 s<sup>-1</sup> (~47% decline) in 2016 and ~4.1 s<sup>-1</sup> (~54% decline) in 2017. However, as the impact of the freshening event subsided, both variables rebounded. In 2018, the annual current speed rose to ~8.4 cm/s, and the annual shear jumped to ~8.6 s<sup>-1</sup>, implying a renewed strengthening of upper ocean currents.

The reduced upper-ocean currents and shear in 2015 and 2016 were accompanied by a significant reduction in turbulent ocean heat flux, as shown in Fig. 4 from (Polyakov., 2020b). It shows that during the winters of 2015 and 2016, around the onset of the freshening event, the divergent heat flux across the halocline decreased from 20 W/m² to 3 W/m². This weakening of oceanic heat flux was reflected in the sea ice concentration observed at all moorings during the following summers of 2016 and 2017 (Fig. 15). At moorings M1<sub>1</sub>, M1<sub>2</sub>, and M1<sub>3</sub>, the seasonal decline from June to September slowed in 2016-2017 compared to 2013-2015 and 2018, with sea ice concentration remaining much higher through late summer than in other years. At moorings M1<sub>4</sub>, M1<sub>5</sub>, and M3, summer SIC remained near 50–70 % in 2016–2017, whereas in other years before and after, it dropped to zero between August and October, highlighting the unusual presence of sea ice in offshore regions during the freshening event. Altogether, these findings provide strong evidence that the 2015–2017 freshening event altered upper-ocean stratification, resulting in reduced oceanic heat fluxes and leading to delayed melt and more summer sea ice, emphasizing the sensitivity of upper-ocean processes and sea ice to episodic freshwater forcing in the Arctic.

Figure 13: Twenty-three months back trajectories calculated from six mooring locations in the eastern Eurasian Basin using ORAS5 current velocity fields (0-5 m) for a month of each period: April 2015 pre-event), April

<sup>419 2016 (</sup>event), and October 2017 post-event). Each color line represents a Lagrangian path traced backward from a mooring location: M1<sub>1</sub> (dark blue), M1<sub>2</sub> (light blue), M1<sub>3</sub> (green), M1<sub>4</sub> (yellow), M1<sub>5</sub> (orange), M3 (red).

Figure 14: Consequences of the 2015-2017 freshening event in the eastern Eurasian Basin. (a) Time-depth section of squared buoyancy frequency  $N^2$  for the  $M1_3$  mooring location (Adapted from Polyakov et al., 2020b-J. Climate); (b,c) Time series of normalized (reduced to anomalies by subtracting means, Mn, and divided by standard deviations, SD) current speed |U|, vertical shear of horizontal current  $|U_z|$  (both from 10m depth level), and sea ice concentration, SIC (the latter time series are multiplied by minus one) at M1 mooring location. Blue lines represent total current speed and shear, while grey lines indicate SIC. Mn and SD are provided for |U| in cm/s,  $|U_z|$  in  $10^3$  s<sup>-1</sup>, and SIC in %. Correlations R between |U| and SIC (blue digits) are statistically significant at the 0.05% level.

Figure 15: Time series of daily sea ice concentration (%) from 2013 to 2018 for six mooring locations in the eastern Eurasian Basin.

| 433 | 4. Discussion and Conclusions                                                                                                           |
|-----|-----------------------------------------------------------------------------------------------------------------------------------------|
| 434 | 4.1 Summary of findings                                                                                                                 |
| 435 | An extreme freshening event, spanning October 2015 – March 2017, was observed at six moorings (M1 <sub>1</sub> –M1 <sub>5</sub> ,       |
| 436 | M3) in the eastern Eurasian Basin, beginning at the shelf moorings M1 <sub>1</sub> –M1 <sub>3</sub> in late 2015 and reaching the       |
| 437 | offshore moorings M1 <sub>4</sub> , M1 <sub>5</sub> , and M3 by early 2016, persisting into early 2017 (Figs. 2 and 3). Salinity in the |
| 438 | upper 175 m decreased by an average of 0.5 psu, which is equivalent to ~0.60 m of freshwater (Fig. 4), and                              |
| 439 | nearly doubling the available potential energy at the shelf moorings, thereby enhancing stratification during the                       |
| 440 | freshening event (Fig. 5). The observed freshening cannot be explained by the cross-slope shifts of the AW salty                        |
| 441 | and warm core (Table S2; Fig. S5). Instead, chemical observations indicate that meteoric (river) water                                  |
| 442 | predominantly contributed to the freshwater content in the Eurasian Basin, doubled from 0.06% in 2013 to                                |
| 443 | 0.12% in 2015 (Fig. 8). Discharge records show an exceptional increase from the Yenisey and Ob rivers in                                |
| 444 | 2014–2015 relative to 2013 (Fig. 9), with Yenisey contributing $\sim$ 0.34m and Ob contributing $\sim$ 0.44m in 2014-                   |
| 445 | 2015 of freshwater when spread over the mooring domain, which together could supply enough freshwater to                                |
| 446 | account for the observed freshwater in the eastern Eurasian Basin. Moreover, low-salinity anomalies detected in                         |
| 447 | the Kara Sea in spring 2015 (Fig. 10) further supported the connection between Siberian River discharge and the                         |
| 448 | freshening observed in the Eurasian Basin. The trajectory analysis reveals that the upper ocean freshening found                        |
| 449 | in the mooring records indeed originated from the Kara Sea, emphasizing the Yenisey and Ob discharge as the                             |
| 450 | primary source. Cross-correlation analysis between time series of Kara and Laptev Sea salinity shows that                               |
| 451 | freshwater transport took about 8-9 months to reach shelf moorings M1 <sub>1</sub> and M1 <sub>2</sub> and about 22-23 months to        |
| 452 | reach offshore moorings $M1_3$ – $M1_5$ and $M3$ (Fig. 11), which aligns with the varying start dates of freshening                     |
| 453 | observed at the moorings (Fig. 3). Although ERA5 winds changed from southeasterly to southwesterly direction                            |
| 454 | during 2015-2017 (Fig. 12), the trajectory pathways remained insensitive to these changes in wind direction.                            |
| 455 | The 2015-2017 freshening increased stratification, which can inhibit vertical heat ventilation from the AW                              |
| 456 | below. Indeed, Polyakov et al., (2020b) showed that the divergent heat flux across the halocline declined during                        |
| 457 | the winters of 2015 and 2016 from approximately 20W/m² to about 3W/m². This diminished upward transfer of                               |
| 458 | heat allows the sea ice to grow thicker. Consequently, thicker ice persisted through the melting summer season                          |
| 459 | and delayed sea-ice melt. As a result, the summer sea ice concentration at offshore moorings in 2016-2017                               |
| 460 | stayed between 50-70%, instead of declining to near-zero as in 2013-2015 and 2018 (Fig. 15). At the same                                |
| 461 | time, during the freshening period, surface current speed and vertical shear in the upper 10m experienced a                             |
| 462 | significant weakening by as much as 90% (Fig. 14).                                                                                      |
| 463 | 4.2 Broader climate implications                                                                                                        |
| 464 | The observed freshening between 2015 and 2017, along with increased upper-ocean stratification, resulted in                             |
| 465 | more stable halocline conditions. This freshwater, therefore, can slow down atlantification (a part of climate                          |
| 466 | change associated with the advection of anomalous water properties from upstream basins), because the buoyant                           |
| 467 | freshwater layer limits the upward ventilation from the Atlantic warm layer (e.g., Polyakov et al., 2017, 2023).                        |
| 468 | Following the 2015-2017 event, Polyakov et al., (2020b) reported an intense release of accumulated subsurface                           |
| 469 | heat in 2018, with vertical heat flux increased by almost a factor of two relative to the previous years, causing                       |
| 470 | extensive sea ice loss. Modeling experiments also demonstrate that adding freshwater can delay ice melt by                              |
| 471 | strengthening stratification (Zhang et al., 2023).                                                                                      |

At much longer time scales, during the Younger Dryas and 8.2 ka events, large freshwater discharges strengthened Arctic stratification and expanded sea-ice cover, yet subsequent re-ventilation of oceanic heat produced rapid warming and long-lasting ice retreat (Fahl and Stein, 2012; Spielhagen and Bauch, 2015). This illustrate that freshwater perturbations can act first as a stabilizing barrier, then as a trigger for abrupt transitions in ocean-ice regimes once the barrier erodes. At present, riverine-driven freshening could operate similarly by alternately insulating and exposing the Arctic's subsurface heat reservoir. Once extensive sea-ice loss occurs, positive feedback such as reduced albedo and enhanced ocean heat uptake reinforce the warming and hinder sea ice recovery (Dörr et al., 2021). Hence, existing observations, model studies, and paleoclimate examples together suggest that freshwater anomalies can exert a powerful control over Arctic stratification, sea-ice dynamics, and climate stability. 4.3 Uncertainties and Limitations This study has a few limitations and uncertainties that should be considered when interpreting the findings. First, the mooring array provides reasonable multi-year coverage in the Eurasian Basin; however, mooring observations start at a depth of ~30 m or greater (see Table 1) to avoid ice keels. Consequently, the maximum freshening that often resides in the very top layer cannot be monitored, and the overall magnitude of freshening may be underestimated by the available mooring records. Vertical resolution varies among moorings: singledepth CTD sensors provide coarser resolution, while MMPs offer continuous vertical profiles. However, for integral characteristics like freshwater content, this had minimal effect (see Methods for details). Another potential limitation comes from the assumption that the parcels remain at a fixed depth in our trajectory experiments using ORAS5 velocities, thus neglecting vertical motions that could alter their travel times or pathways. However, the freshwater pathway from the Kara Sea to the Laptev Sea identified in our analysis is well supported by observations (Janout et al., 2015; Osadchiev et al., 2023), reinforcing confidence in our result. Also, while ORAS5 performs better than several other reanalyses and model products in the Arctic (Hall et al., 2022), it still tends to overestimate sea-surface salinity and under-resolve currents in ice-covered regions (Jin et al., 2023). We used this reanalysis-based Kara Sea salinity data in the spatial distributions of salinities (Fig. 10) and for correlation analysis (Fig. 11). This bias may impact estimates of salinity in the Kara Sea, potentially reducing the observed correlations between the Kara and Laptev Sea time series. Enhanced observations in the surface ocean layer and more sophisticated data assimilation in reanalysis models will help reduce uncertainties related to the above shortcomings. Despite these limitations, the core findings, including the anomalous freshening of the eastern Eurasian Basin and northern Laptev Sea driven by Yenisey and Ob dominated river discharge, and its impacts on sea ice and upper-ocean currents, are robust and well supported by extensive observations. 4.4. Final note Understanding the role of episodic events is essential for assessing their influence on upper-ocean stratification, sea ice variability, and potential feedback within the Arctic climate system. This underscores the need for long-term sustained mooring deployments, with enhanced observations in the near-surface layer, and integration of these datasets with reanalysis to better understand and predict the cascading consequences of freshening events on Arctic climate and ecosystems. Our results demonstrate that extreme river discharge events

## https://doi.org/10.5194/egusphere-2025-5592 Preprint. Discussion started: 19 November 2025 © Author(s) 2025. CC BY 4.0 License.

| 511 | on regional sea-ice cover. Improved representation of such episodic events in coupled models will be essential  |
|-----|-----------------------------------------------------------------------------------------------------------------|
| 512 | for predicting future Arctic climate variability.                                                               |
|     |                                                                                                                 |
| 513 | Acknowledgements. IVP acknowledges funding from Office of Naval Research Grant N00014-21-1-2577. DM             |
| 514 | and IVP were supported by National Science Foundation (NSF) grant #1724523 and the U.S. Department of           |
| 515 | Energy grant 280253.                                                                                            |
| 516 | Competing interests: The authors declare that they have no conflict of interest.                                |
| 517 | Author Contributions. All authors participated in preliminary analysis, data processing and analysis of mooring |
| 518 | data, interpretation of hydrographic data and formulating objectives of the study. All authors contributed to   |
| 519 | interpreting the data and writing the paper.                                                                    |
| 520 | Author Information. Correspondence and requests should be addressed to IVP (ivpolyakov@alaska.edu).             |
| 521 | Data Availability Statement. All mooring data used in this study are available at                               |
| 522 | https://arcticdata.io/catalog/#view/arctic-data. The ERA5 reanalysis data is available from                     |
| 523 | https://cds.climate.copernicus.eu/cdsapp#!/home. Sea ice concentration is available from                        |
| 524 | https://www.ncdc.noaa.gov/oisst.                                                                                |
| 525 |                                                                                                                 |

| 526        | References                                                                                                                                                                     |
|------------|--------------------------------------------------------------------------------------------------------------------------------------------------------------------------------|
| 527        | Aagaard, K. and Carmack, E. C.: The role of sea ice and other fresh water in the Arctic circulation, Journal of                                                                |
| 528        | Geophysical Research: Oceans, 94, 14485–14498, https://doi.org/10.1029/JC094iC10p14485, 1989.                                                                                  |
| 529        | Aksenov, Y., Ivanov, V. V., Nurser, A. J. G., Bacon, S., Polyakov, I. V., Coward, A. C., Naveira-Garabato, A.                                                                  |
| 530        | C., and Beszczynska-Moeller, A.: The arctic circumpolar boundary current, Journal of Geophysical                                                                               |
| 531        | Research: Oceans, 116, https://doi.org/10.1029/2010JC006637, 2011.                                                                                                             |
| 532        | Alkire, M. B., Morison, J., Schweiger, A., Zhang, J., Steele, M., Peralta-Ferriz, C., and Dickinson, S.: A                                                                     |
| 533        | Meteoric Water Budget for the Arctic Ocean, Journal of Geophysical Research: Oceans, 122, 10020-                                                                               |
| 534        | 10041, https://doi.org/10.1002/2017JC012807, 2017.                                                                                                                             |
| 535        | Bauch, D., Schlosser, P., and Fairbanks, R. G.: Freshwater balance and the sources of deep and bottom waters in                                                                |
| 536        | the Arctic Ocean inferred from the distribution of H218O, Progress in Oceanography, 35, 53-80,                                                                                 |
| 537        | https://doi.org/10.1016/0079-6611(95)00005-2, 1995.                                                                                                                            |
| 538        | Bauch, D., Hölemann, J. A., Nikulina, A., Wegner, C., Janout, M. A., Timokhov, L. A., and Kassens, H.:                                                                         |
| 539        | Correlation of river water and local sea-ice melting on the Laptev Sea shelf (Siberian Arctic), Journal of                                                                     |
| 540        | Geophysical Research: Oceans, 118, 550–561, https://doi.org/10.1002/jgrc.20076, 2013.                                                                                          |
| 541        | Baumann, T. M., Polyakov, I. V., Pnyushkov, A. V., Rember, R., Ivanov, V. V., Alkire, M. B., Goszczko, I.,                                                                     |
| 542        | and Carmack, E. C.: On the seasonal cycles observed at the continental slope of the Eastern Eurasian basin                                                                     |
| 543<br>544 | of the Arctic Ocean, Journal of Physical Oceanography, 48, 1451–1470, https://doi.org/10.1175/JPO-D-                                                                           |
| 545        | 17-0163.1, 2018. Carmack, E., Polyakov, I., Padman, L., Fer, I., Hunke, E., Hutchings, J., Jackson, J., Kelley, D., Kwok, R.,                                                  |
| 546        | Layton, C., Melling, H., Perovich, D., Persson, O., Ruddick, B., Timmermans, ML., Toole, J., Ross, T.,                                                                         |
| 547        | Vavrus, S., and Winsor, P.: Toward Quantifying the Increasing Role of Oceanic Heat in Sea Ice Loss in                                                                          |
| 548        | the New Arctic, Bulletin of the American Meteorological Society, https://doi.org/10.1175/BAMS-D-13-                                                                            |
| 549        | 00177.1, 2015.                                                                                                                                                                 |
| 550        | Cavalieri, D. J., Parkinson, C. L., and Vinnikov, K. Y.: 30-Year satellite record reveals contrasting Arctic and                                                               |
| 551        | Antarctic decadal sea ice variability, Geophysical Research Letters, 30,                                                                                                       |
| 552        | https://doi.org/10.1029/2003GL018031, 2003.                                                                                                                                    |
| 553        | Comiso, J.: Bootstrap Sea Ice Concentrations from Nimbus-7 SMMR and DMSP SSM/I-SSMIS, Version 3,                                                                               |
| 554        | https://doi.org/10.5067/7Q8HCCWS4I0R, 2017.                                                                                                                                    |
| 555        | Dai, A. and Trenberth, K. E.: Estimates of Freshwater Discharge from Continents: Latitudinal and Seasonal                                                                      |
| 556        | Variations, J. Hydrometeor., 3, 660–687, https://doi.org/10.1175/1525-                                                                                                         |
| 557        | 7541(2002)003%3C0660:EOFDFC%3E2.0.CO;2, 2002.                                                                                                                                  |
| 558        | Dittmar, T. and Kattner, G.: The biogeochemistry of the river and shelf ecosystem of the Arctic Ocean: a                                                                       |
| 559        | review, Marine Chemistry, 83, 103–120, https://doi.org/10.1016/S0304-4203(03)00105-1, 2003.                                                                                    |
| 560<br>561 | Dörr, J., Årthun, M., Eldevik, T., and Madonna, E.: Mechanisms of Regional Winter Sea-Ice Variability in a                                                                     |
| 562        | Warming Arctic, https://doi.org/10.1175/JCLI-D-21-0149.1, 2021. Fahl, K. and Stein, R.: Modern seasonal variability and deglacial/Holocene change of central Arctic Ocean sea- |
| 563        | ice cover: New insights from biomarker proxy records, Earth and Planetary Science Letters, 351–352,                                                                            |
| 564        | 123–133, https://doi.org/10.1016/j.epsl.2012.07.009, 2012.                                                                                                                     |
| 565        | Haine, T. W. N., Curry, B., Gerdes, R., Hansen, E., Karcher, M., Lee, C., Rudels, B., Spreen, G., de Steur, L.,                                                                |
| 566        | Stewart, K. D., and Woodgate, R.: Arctic freshwater export: Status, mechanisms, and prospects,                                                                                 |
| 567        | EPIC3Global and Planetary Change, 125, pp. 13-35, ISSN: 09218181, 125, 13–35,                                                                                                  |
| 568        | https://doi.org/10.1016/J.GLOPLACHA.2014.11.013, 2015.                                                                                                                         |
| 569        | Hall, S. B., Subrahmanyam, B., and Morison, J. H.: Intercomparison of Salinity Products in the Beaufort Gyre                                                                   |
| 570        | and Arctic Ocean, Remote Sensing, 14, 71, https://doi.org/10.3390/rs14010071, 2022.                                                                                            |
| 571        | Hersbach, H., Bell, B., Berrisford, P., Hirahara, S., Horányi, A., Muñoz-Sabater, J., Nicolas, J., Peubey, C.,                                                                 |
| 572        | Radu, R., Schepers, D., Simmons, A., Soci, C., Abdalla, S., Abellan, X., Balsamo, G., Bechtold, P.,                                                                            |
| 573        | Biavati, G., Bidlot, J., Bonavita, M., De Chiara, G., Dahlgren, P., Dee, D., Diamantakis, M., Dragani, R.,                                                                     |
| 574        | Flemming, J., Forbes, R., Fuentes, M., Geer, A., Haimberger, L., Healy, S., Hogan, R. J., Hólm, E.,                                                                            |
| 575        | Janisková, M., Keeley, S., Laloyaux, P., Lopez, P., Lupu, C., Radnoti, G., de Rosnay, P., Rozum, I.,                                                                           |
| 576        | Vamborg, F., Villaume, S., and Thépaut, JN.: The ERA5 global reanalysis, Quarterly Journal of the                                                                              |
| 577        | Royal Meteorological Society, 146, 1999–2049, https://doi.org/10.1002/qj.3803, 2020.                                                                                           |
| 578<br>579 | Hoffman, E. L., Subrahmanyam, B., Trott, C. B., and Hall, S. B.: Comparison of Freshwater Content and                                                                          |
| 580        | Variability in the Arctic Ocean Using Observations and Model Simulations, Remote Sensing, 15, 3715, https://doi.org/10.3390/rs15153715.2023                                    |
| 581        | https://doi.org/10.3390/rs15153715, 2023.  Holmes, R. M., McClelland, J., Tank, S., Spencer, R., and Shiklomanov, A.: Arctic Great Rivers Observatory                          |
| 582        | IV Biogeochemistry and Discharge Data: 2020-2024, https://doi.org/10.18739/A2FQ9Q683, 2022.                                                                                    |
| 583        | IPPC: Climate Change 2021: The Physical Science Basis. Contribution of Working Group I to the Sixth                                                                            |
| 584        | Assessment Report of the Intergovernmental Panel on Climate Change,                                                                                                            |
| 585        | https://doi.org/10.1017/9781000157896.2021                                                                                                                                     |

597

610

611

612

613

620

- Janout, M. A., Aksenov, Y., Hölemann, J. A., Rabe, B., Schauer, U., Polyakov, I. V., Bacon, S., Coward, A. C.,
   Karcher, M., Lenn, Y. D., Kassens, H., and Timokhov, L.: Kara Sea freshwater transport through Vilkitsky
   Strait: Variability, forcing, and further pathways toward the western Arctic Ocean from a model and
   observations, Journal of Geophysical Research: Oceans, 120, 4925–4944,
   https://doi.org/10.1002/2014JC010635, 2015.
- Jin, Y., Chen, M., Yan, H., Wang, T., and Yang, J.: Sea level variation in the Arctic Ocean since 1979 based on
   ORAS5 data, Front. Mar. Sci., 10, https://doi.org/10.3389/fmars.2023.1197456, 2023.
   Kim, B. M., Hong, J. Y., Jun, S. Y., Zhang, X., Kwon, H., Kim, S. J., Kim, J. H., Kim, S. W., and Kim, H. K.:
  - Kim, B. M., Hong, J. Y., Jun, S. Y., Zhang, X., Kwon, H., Kim, S. J., Kim, J. H., Kim, S. W., and Kim, H. K.: Major cause of unprecedented Arctic warming in January 2016: Critical role of an Atlantic windstorm, Scientific Reports, 7, 1–9, https://doi.org/10.1038/srep40051, 2017.
  - Laukert, G., Bauch, D., Rabe, B., Krumpen, T., Damm, E., Kienast, M., Hathorne, E., Vredenborg, M., Tippenhauer, S., Andersen, N., Meyer, H., Mellat, M., D'Angelo, A., Simões Pereira, P., Nomura, D., Horner, T. J., Hendry, K., and Kienast, S. S.: Dynamic ice—ocean pathways along the Transpolar Drift amplify the dispersal of Siberian matter, Nat Commun, 16, 3172, https://doi.org/10.1038/s41467-025-57881-9, 2025.
  - Miller, J. R.: The Salinity Effect in a Mixed Layer Ocean Model, Journal of Physical Oceanography, 6, 29–35, https://doi.org/10.1175/1520-0485(1976)006%3C0029:TSEIAM%3E2.0.CO;2, 1976.
    - Müller, V., Wang, Q., Koldunov, N., Danilov, S., Sidorenko, D., and Jung, T.: Variability of Eddy Kinetic Energy in the Eurasian Basin of the Arctic Ocean Inferred From a Model Simulation at 1-km Resolution, Journal of Geophysical Research: Oceans, 129, e2023JC020139, https://doi.org/10.1029/2023JC020139, 2024.
- Nguyen, A. T., Menemenlis, D., and Kwok, R.: Arctic ice-ocean simulation with optimized model parameters:

  Approach and assessment, Journal of Geophysical Research: Oceans, 116,

  https://doi.org/10.1029/2010JC006573, 2011.
  - Osadchiev, A., Sedakov, R., Frey, D., Gordey, A., Rogozhin, V., Zabudkina, Z., Spivak, E., Kuskova, E., Sazhin, A., and Semiletov, I.: Intense zonal freshwater transport in the Eurasian Arctic during ice-covered season revealed by in situ measurements, Sci Rep, 13, 16508, https://doi.org/10.1038/s41598-023-43524-w, 2023.
- Osadchiev, A., Kuskova, E., and Ivanov, V.: The roles of river discharge and sea ice melting in formation of freshened surface layers in the Kara, Laptev, and East Siberian seas, Front. Mar. Sci., 11, https://doi.org/10.3389/fmars.2024.1348450, 2024.
- Overland, J. E. and Wang, M.: Recent Extreme Arctic Temperatures are due to a Split Polar Vortex,
   https://doi.org/10.1175/JCLI-D-16-0320.1, 2016.
   Pemberton, P., Nilsson, J., and Meier, H. E. M.: Arctic Ocean freshwater composition, pathways and
  - Pemberton, P., Nilsson, J., and Meier, H. E. M.: Arctic Ocean freshwater composition, pathways and transformations from a passive tracer simulation, Tellus A: Dynamic Meteorology and Oceanography, 66, 23988, https://doi.org/10.3402/tellusa.v66.23988, 2014.
  - Pnyushkov, A. V., Polyakov, I. V., Ivanov, V. V., Aksenov, Y., Coward, A. C., Janout, M., and Rabe, B.: Structure and variability of the boundary current in the Eurasian Basin of the Arctic Ocean, Deep Sea Research Part I: Oceanographic Research Papers, 101, 80–97, https://doi.org/10.1016/j.dsr.2015.03.001, 2015.
  - Polyakov, I. V., Pnyushkov, A. V., Rember, R., Padman, L., Carmack, E. C., and Jackson, J. M.: Winter Convection Transports Atlantic Water Heat to the Surface Layer in the Eastern Arctic Ocean, Journal of Physical Oceanography, 43, 2142–2155, https://doi.org/10.1175/JPO-D-12-0169.1, 2013.
  - Polyakov, I. V., Pnyushkov, A. V., Alkire, M. B., Ashik, I. M., Baumann, T. M., Carmack, E. C., Goszczko, I., Guthrie, J., Ivanov, V. V., Kanzow, T., Krishfield, R., Kwok, R., Sundfjord, A., Morison, J., Rember, R., and Yulin, A.: Greater role for Atlantic inflows on sea-ice loss in the Eurasian Basin of the Arctic Ocean, Science, 356, 285–291, https://doi.org/10.1126/science.aai8204, 2017.
  - Polyakov, I. V., Pnyushkov, A. V., and Carmack, E. C.: Stability of the arctic halocline: a new indicator of arctic climate change, Environmental Research Letters, 13, 125008, https://doi.org/10.1088/1748-9326/aaec1e, 2018.
- Polyakov, I. V., Rippeth, T. P., Fer, I., Baumann, T. M., Carmack, E. C., Ivanov, V. V., Janout, M., Padman, L.,
   Pnyushkov, A. V., and Rember, R.: Intensification of Near-Surface Currents and Shear in the Eastern
   Arctic Ocean, Geophysical Research Letters, 47, 1–9, https://doi.org/10.1029/2020GL089469, 2020a.
- Polyakov, I. V., Rippeth, T. P., Fer, I., Alkire, M. B., Baumann, T. M., Carmack, E. C., Ingvaldsen, R., Ivanov,
  V. V., Janout, M., Lind, S., Padman, L., Pnyushkov, A. V., and Rember, R.: Weakening of cold halocline
  layer exposes sea ice to oceanic heat in the eastern arctic ocean, Journal of Climate, 33, 8107–8123,
  https://doi.org/10.1175/JCLI-D-19-0976.1, 2020b.
- Polyakov, I. V., Ingvaldsen, R. B., Pnyushkov, A. V., Bhatt, U. S., Francis, J. A., Janout, M., Kwok, R., and
   Skagseth, Ø.: Fluctuating Atlantic inflows modulate Arctic atlantification, Science, 381, 972–979,
   https://doi.org/10.1126/science.adh5158, 2023.

675

676

677

684

685

686

687

- Polyakov, I. V., Pnyushkov, A. V., Carmack, E. C., Charette, M., Cho, K.-H., Dykstra, S., Haapala, J., Jung, J., 647 Kipp, L., and Yang, E. J.: Role of sea ice, stratification, and near-inertial oscillations in shaping the upper 648 Siberian Arctic Ocean currents, EGUsphere, 1-27, https://doi.org/10.5194/egusphere-2025-2316, 2025. 649
  - Rawlins, M. A. and Karmalkar, A. V.: Regime shifts in Arctic terrestrial hydrology manifested from impacts of climate warming, The Cryosphere, 18, 1033-1052, https://doi.org/10.5194/tc-18-1033-2024, 2024.
- Rawlins, M. A., Steele, M., Holland, M. M., Adam, J. C., Cherry, J. E., Francis, J. A., Groisman, P. Y., 652 Hinzman, L. D., Huntington, T. G., Kane, D. L., Kimball, J. S., Kwok, R., Lammers, R. B., Lee, C. M., 653 Lettenmaier, D. P., McDonald, K. C., Podest, E., Pundsack, J. W., Rudels, B., Serreze, M. C., 654 Shiklomanov, A., Skagseth, Ø., Troy, T. J., Vörösmarty, C. J., Wensnahan, M., Wood, E. F., Woodgate, 655 R., Yang, D., Zhang, K., and Zhang, T.: Analysis of the Arctic System for Freshwater Cycle 656 Intensification: Observations and Expectations, https://doi.org/10.1175/2010JCLI3421.1, 2010.
  - Serreze, M. C., Barrett, A. P., Slater, A. G., Woodgate, R. A., Aagaard, K., Lammers, R. B., Steele, M., Moritz, R., Meredith, M., and Lee, C. M.: The large-scale freshwater cycle of the Arctic, Journal of Geophysical Research: Oceans, 111, https://doi.org/10.1029/2005JC003424, 2006.
  - Shiklomanov, A., Déry, S., Tretiakov, M., Yang, D., Magritsky, D., Georgiadi, A., and Tang, W.: River Freshwater Flux to the Arctic Ocean, 703–738, https://doi.org/10.1007/978-3-030-50930-9\_24, 2021.
    - Shiklomanov, A. I., Yakovleva, T. I., Lammers, R. B., Karasev, I. Ph., Vörösmarty, Charles. J., and Linder, E.: Cold region river discharge uncertainty—estimates from large Russian rivers, Journal of Hydrology, 326, 231–256, https://doi.org/10.1016/j.jhydrol.2005.10.037, 2006.
    - Spielhagen, R. F. and Bauch, H. A.: The role of Arctic Ocean freshwater during the past 200 ky, Arktos, 1, 18, https://doi.org/10.1007/s41063-015-0013-9, 2015.
    - Stadnyk, T. A., Tefs, A., Broesky, M., Déry, S. J., Myers, P. G., Ridenour, N. A., Koenig, K., Vonderbank, L., and Gustafsson, D.: Changing freshwater contributions to the Arctic: A 90-year trend analysis (1981-2070), Elementa: Science of the Anthropocene, 9, 00098, https://doi.org/10.1525/elementa.2020.00098,
- Stroeve, J. C., Markus, T., Boisvert, L., Miller, J., and Barrett, A.: Changes in Arctic melt season and 672 implications for sea ice loss, Geophysical Research Letters, 41, 1216-1225, 673 https://doi.org/10.1002/2013GL058951, 2014. 674
  - Thurnherr, A. M., Goszczko, I., and Bahr, F.: Improving LADCP Velocity with External Heading, Pitch, and Roll, https://doi.org/10.1175/JTECH-D-16-0258.1, 2017.
  - Timmermans, M.-L. and Marshall, J.: Understanding Arctic Ocean Circulation: A Review of Ocean Dynamics in a Changing Climate, Journal of Geophysical Research: Oceans, 125, e2018JC014378, https://doi.org/10.1029/2018JC014378, 2020.
- 679 Torrence, C. and Compo, G. P.: A Practical Guide to Wavelet Analysis, 1998.
- Whitmore, L., Kaufman, M., Pnuyshkov, A., and Polyakov, I.: Nansen and Amundsen Basins Observational 681 System II (NABOS II) - Seawater macronutrient observations in the eastern Eurasian and Makarov 682 Basins, Arctic Ocean, 2021, https://doi.org/10.18739/A25T3G17R, 2023. 683
  - Yamamoto-Kawai, M., McLaughlin, F., and Carmack, E.: Ocean acidification in the three oceans surrounding northern North America, Journal of Geophysical Research: Oceans, 118, 6274-6284, https://doi.org/10.1002/2013JC009157, 2013.
  - Zhang, H., Bai, X., and Wang, K.: Response of the Arctic sea ice-ocean system to meltwater perturbations based on a one-dimensional model study, Ocean Science, 19, 1649-1668, https://doi.org/10.5194/os-19-1649-2023, 2023.
- Zuo, H., Balmaseda, M. A., Tietsche, S., Mogensen, K., and Mayer, M.: The ECMWF operational ensemble 690 reanalysis-analysis system for ocean and sea ice: a description of the system and assessment, Ocean 691 Science, 15, 779-808, https://doi.org/10.5194/os-15-779-2019, 2019. 692