# Peer review of "1. Introduction"

_EGUsphere, 2025_

## Referee Comment (RC1)

*Review for*

**Documenting the 2015-2017 freshening of the eastern Eurasian Basin of the Arctic Ocean and evaluating its drivers and consequences**

**by Dolly More and Igor V. Polyakov**

General comments:

This manuscript by More and Polyakov documents a freshening event that occurred in the Eastern Eurasian Basin between 2015 and 2017, based on observations from several moorings deployed in the region. The authors characterize this anomaly through estimates of freshwater content and available potential energy at the mooring sites. While the absence of near-surface observations necessarily limits the precision of these estimates, the dataset itself is valuable and the event is potentially important.

The authors propose that the freshening is primarily driven by increased discharge from the Yenisey and Ob rivers in the years preceding the event. This interpretation is supported by a lagged correlation analysis using ORAS5 surface salinity, Lagrangian trajectory modeling, and estimates of the meteoric water fraction derived from in situ observations. The manuscript further discusses possible consequences of the freshening, suggesting enhanced stratification, reduced vertical mixing, weakened surface currents, and reduced sea ice melt during the event.

While the freshening event and the observational dataset are of clear interest, I find that the current analysis remains somewhat limited in depth, and that several conclusions appear stronger than what is fully supported by the presented evidence at this stage. In addition, some arguments would benefit from clarification or revision, as certain interpretations appear inconsistent or insufficiently justified.

The overall quality of the figures is also a significant concern. Several figures lack axis labels, have readability issues, or are difficult to interpret in their current form. Substantial effort is needed to improve figure clarity and consistency before the manuscript can be considered publication-ready. In addition, some figures appear redundant and do not always add new information. For instance, Figures 2–7 show overlapping aspects of the same signal, Figure 8 is extremely difficult to read, Figure 12 adds limited insight, and Figure 15 duplicates information already shown in Figure 14. Some of these figures could likely be removed or moved to the Supplementary Materials.

From a scientific perspective, the proposed link between river discharge and the observed freshening is likely, but several aspects of the observed signal remain unexplained and would benefit from discussion. In particular:

- What is happening in the surface layer, and is it possible that the anomaly was already present there in summer 2015?
- Why does the freshwater anomaly appear to arrive abruptly down to ~100 m at some moorings, while the meteoric water fraction increase is confined to the upper ~25 m (Fig. 8)?

- Why does the anomaly appear weaker in summer 2016 compared to the preceding and following winters?
- Why is the freshwater anomaly associated with different (and sometimes opposing) temperature signals at different moorings?

While definitive answers to all of these questions may not be possible, acknowledging and discussing these features would improve the physical interpretation of the results.

The discussion of the impacts of the freshening event would also benefit from clarification. In particular, the link between freshening and changes in surface currents is not entirely clear, and it is difficult to assess whether the freshening alone can explain the increased sea-ice cover observed in summers 2016 and 2017, given the potential influence of other factors.

Finally, some aspects of the analysis are unclear. For example, the calculation of $\overline{V}.\Delta S$ does not seem to provide information about a potential cross-slope shift of the Atlantic Water core, as it compares a mean-state salt flux with a temporal salinity anomaly. This interpretation likely needs to be reconsidered.

Specific comments:

**l. 32–34:** "*exemplifying the phenomenon of Arctic amplification*"
The logical connection in this sentence is unclear. The numbers in the paragraph show that the Arctic is warming, but they do not illustrate Arctic amplification.

**l. 47–53:** Consider moving this to l. 38, where you introduce the presence of freshwater in the Arctic.

**l. 90–93:** "*indicating that interpolation of SBE data introduces negligible error.*"
You have not yet mentioned that the data are interpolated. I suggest moving this sentence to the end of the subsection.

**l. 149:** Is this wavelet-based method more accurate than simply analyzing anomalies relative to the seasonal cycle?

**l. 159–170:** I would shorten this paragraph: simply define z1 and z2, and state that using the depth at which the anomaly becomes statistically indistinguishable from zero would yield similar results.

**Definitions of freshwater content, Q, and APE:** Is z1 the surface or the depth of the shallowest reliable salinity record zl defined earlier? You state that it is the surface for APE, but this is not clear for freshwater content and Q.

**Analysis of Figure 2:** The abrupt arrival of the fresh anomaly at moorings M14 and M3 in January 2016 is intriguing. The anomaly then weakens during the following summer before strengthening again in winter 2016–2017. This variability is also visible in Figures 4, 5 and 6. Why do the anomalies appear stronger during the two winters? Is it because part of the freshwater anomaly resides closer to the surface in summer (in a layer not resolved by the

moorings) and is transferred to deeper layers by winter mixing? Or is there another explanation?

**l. 267:** "*For instance, the freshwater content averaged over four deeper moorings increased by 0.67 m.*"
The statement that freshwater content increased by 0.67 m when averaged over four deeper moorings is difficult to interpret meaningfully, given surface-layer limitations (see also l. 330–332).

**Figures 3 and 4:** There is some redundancy between these figures; choosing one would likely be sufficient. The qualitative results (lower salinity corresponding to larger freshwater content and higher APE) are obvious, and quantitative estimates are not possible because the moorings do not cover the surface layer.

**l. 281–288:** The contrasting tendencies in temperature and OHC between the moorings are difficult to interpret. At mooring M11, for instance, the average anomalies have opposite signs (positive temperature anomaly and negative salinity anomaly), yet temperature (or OHC) and salinity (or −FWC) appear correlated during 2015–2017. Do you have an explanation for this?
You also state that "*The potential underlying mechanisms for these contrasting signals are discussed in Section 6,*" but there is no Section 6 in the manuscript. Such a discussion would improve the interpretation of the results.

**Figure 7:** The figure is difficult to read and would benefit from improved resolution.

**l. 297–310:** This analysis is unclear to me. As I understand it, $\overline{V}.\Delta S$ is calculated from time-averaged velocity and salinity at each mooring location, representing one component of the mean salt balance. You then compare it with salinity anomalies during the freshening event, which represent temporal changes. I therefore do not see why these two quantities should be related. One possible alternative might be to examine changes in $\overline{V}.\Delta S$ between the pre-2015 period and the freshening event.

**Table S2:** Please add units to the column headers, or remove the table if you agree with my previous comment on this calculation.

**Figure 8:** This figure is difficult to read. It would benefit from improved resolution, enlarged caption, and axis titles and units. Also, it would be good to specify the months over which these profiles were computed. This may be important for your analysis if the increase in meteoric water fraction was already present in summer 2015.
From what I can see, the main changes in meteoric water occur in the upper 25 m, which are not covered by the moorings. It would be useful to propose an explanation. Could this initially be a surface anomaly starting in summer 2015 that is not detected by the moorings and is later transferred to deeper layers by winter mixing?

**l. 326:** It is not clear from the figure itself that Yenisey discharge was lower in 2013 than in subsequent years, the calculations provided at l. 327-329 are necessary to say that.

**l. 327–330:** Are these anomalies calculated relative to 2013 only? Why not use the 2013–2018 average as a baseline instead?

Also, it is not clear at this stage of the article why the Lena River is not discussed, while it is also shown in Figure 9.

**l. 330–332:** The estimate of 0.60 m of additional freshwater content is difficult to interpret, given both spatial variability among moorings and incomplete vertical coverage. Besides, summing the anomalous discharges from 2013 and 2014 implicitly assumes that freshwater from both years arrives at the mooring line simultaneously, which may not be the case.

**l. 342:** "*The downstream impact of this anomalous freshwater input is evident…*"
The downstream propagation of a single freshwater anomaly is not obvious from the presented maps, which show a mix of local and basin-scale patterns, including local freshening near river discharge regions and broader large-scale anomalies extending into the Eurasian Basin. I would suggest either showing only one or two maps illustrating the spatial extent of the anomaly, or presenting a lagged regression map of surface salinity anomalies onto Yenisey and Ob discharge anomalies.

**Figure 11:** This figure is useful for illustrating the link between river discharge and the freshening anomaly. You state that all correlations are significant at p<0.001. What statistical test was used, and how were the degrees of freedom estimated? I would expect the correlation with M11 (R = 0.39) to be significant at p<0.05, but probably not much more.

**Figure 12:** Captions and titles are cropped, and overall presentation needs improvement.

**l. 365–372:** This explanation is unclear. Winds during the event appear to drive Ekman transport that is along-slope or even toward the coast, whereas the moorings are located in the basin interior. Such winds would be expected to limit freshwater transport toward the moorings.
While it is likely that winds influenced shelf–basin freshwater exchange, it might be more appropriate to focus on wind patterns during the months surrounding the onset of the event (summer–fall 2015) rather than over 2015–2017 as a whole.

**l. 381–382:** "*the resulting trajectories reveal that upper eastern Eurasian Basin freshening originates in the Kara Sea.*" This argument should appear much earlier. All analyses from Figure 9 onward rely on the assumption that the anomaly originates in the Kara Sea.

**Figure 14:** The font, style, and overall appearance differ from the other figures. Please ensure a consistent visual style. The time-axis ticks in panel (a) are not visible.

**l. 395–397:** Are you suggesting that the freshening caused the decrease in surface currents? If so, what physical mechanism would explain this?

**l. 403–405:** "*the divergent heat flux across the halocline decreased from 20 W/m² to 3 W/m².*"
Where do these values come from? Are they taken from Polyakov et al. (2020b)? Do they apply to a specific mooring or represent an average?

**l. 408–414:** You argue that the freshening led to increased sea-ice cover in summers 2016 and 2017, but Figures 2, 4, 5, and 7 suggest that the freshening event ended in spring 2017. It is therefore not obvious how it would affect sea ice during the following summer. Is this due

to delayed effects of reduced winter mixing in 2016–2017? In any case, I would moderate the strength of the claimed link.

**Figure 15:** This figure does not add new information (the SIC time series at M1 already appears in Figure 14). I suggest removing it.

**l. 435:** Why describe this event as "extreme"? It is clearly significant, but no evidence is provided that such events are rare.

**l. 440–441:** "*the observed freshening cannot be explained by cross-slope shifts of the AW salty and warm core.*"
The demonstration of this point is unclear to me.

**l. 443:** "*exceptional increase.*"
The data show an increase, but the time series are too short to justify the term "exceptional."

**l. 446–448:** The timing is not fully consistent: if Kara Sea anomalies in spring 2015 take nearly two years to reach offshore moorings, they would arrive in spring 2017, which you identify as the end of the event.

**l. 452–453:** "… which aligns with the varying start dates of freshening observed at the moorings (Fig. 3)." I would nuance this statement. Differences in the onset of the freshening event among moorings are on the order of 3–4 months, not more than a year.

**l. 456–457:** "Indeed, Polyakov et al., (2020b) showed that the divergent heat flux…"
This was already mentioned and is a result from another paper. I suggest removing it.

**l. 485–487:** "Consequently, the maximum freshening that often resides in the very top layer cannot be monitored, and the overall magnitude of freshening may be underestimated by the available mooring records."
This is an important limitation. Given it, I would place less emphasis on quantitative estimates of freshwater content and APE. An alternative approach could be to analyze the freshening in the ORAS5 reanalysis to estimate the fraction of the anomaly captured by the moorings versus that residing near the surface.

Technical corrections:

**l. 206:** Remove one parenthesis.

**Figure 2:** The caption does not fully correspond to the figure. In particular, there is a confusion between solid and dashed red lines and between red and black lines delimiting the event and the preceding and following periods. The black dashed lines separating the different years could also be mentioned.

**l. 403:** Remove parentheses around the reference.